# Effects of Hypochlorous Acid and Hydrogen Peroxide Treatment on Bacterial Disinfection Treatments in Implantoplasty Procedures

**DOI:** 10.3390/ma16082953

**Published:** 2023-04-07

**Authors:** Esteban Padulles-Gaspar, Esteban Padulles-Roig, Guillermo Cabanes, Román A. Pérez, Javier Gil, Begoña M. Bosch

**Affiliations:** 1Facultad de Odontología, Universitat Internacional de Catalunya, C/Josep Trueta s/n, 08195 Barcelona, Spain; 2Department of Implantology, University of La Salle, Madrid, EDE, C7Gaminedes 11, 28023 Madrid, Spain; 3Bioengineering Institute of Technology, Universitat Internacional de Catalunya, 08195 Barcelona, Spain

**Keywords:** hypochlorous acid, hydrogen peroxide, implantoplasty, periimplantitis

## Abstract

One of the main problems in oral implantology today is peri-implantitis, which affects almost 20% of dental implants placed in patients. One of the most commonly used techniques to eliminate bacterial biofilm is the implantoplasty, that consists of the mechanical modification of the implant surface topography followed by treatments with chemical reagents for decontamination. In this study, the main aim is to evaluate the use of two different chemical treatments based on hypochlorous acid (HClO) and hydrogen peroxide (H_2_O_2_). For this purpose, 75 titanium grade 3 discs were treated with implantoplasty according to established protocols. Twenty-five discs were used as controls, 25 were treated with concentrated HClO and 25 were treated with concentrated HClO followed by treatment with 6% H_2_O_2_. The roughness of the discs was determined using the interferometric process. Cytotoxicity with SaOs-2 osteoblastic cells was quantified at 24 and 72 h, whereas bacteria proliferation using *S. gordonii* and *S. oralis* bacteria was quantified at 5 s and 1 min of treatment. The results showed an increase in the roughness values, the control discs had an Ra of 0.33 μm and those treated with HClO and H_2_O_2_ reached 0.68 μm. Cytotoxicity was present at 72 h, together with a significant proliferation of bacteria. These biological and microbiological results can be attributed to the roughness produced by the chemical agents that triggered bacterial adsorption while inhibiting osteoblast adhesion. The results indicate that even if this treatment can decontaminate the titanium surface after implantation, the produced topography will generate an environment that will not favor long-term performance.

## 1. Introduction

Peri-implantitis is a plaque-associated pathological condition occurring in tissues surrounding dental implants, characterized by inflammation in the peri-implant mucosa and subsequent progressive loss of supporting bone. Peri-implantitis regions exhibit clinical signs of inflammation, bleeding on probing, and/or suppuration, increased probing depths and/or recession of the mucosal margin, in addition to radiographic bone loss [1,2], resulting in loss of supporting structures [3]. Although a host inflammatory responses aiming to activate the immune functions and control the microbial growth are present, this process can also result in a multidrug-tolerant biofilm [4]. Such biofilms are challenging to treat due to their complex microbial composition, their establishment on a favorable substrate, such as Ti, tridimensional structure and increased resistance to antibiotics [5,6]. Several clinical approaches have been studied in order to mitigate their effects, mainly consisting of access, resective and regenerative surgeries, or a combination of these [7,8,9,10,11,12]. following resective surgery, the rough surface of the implant may be exposed to the oral environment. In these cases, implantoplasty has been applied to clean the supracrestal part of the dental implant [8]. Implantoplasty consists of mechanically modifying the implant surface topography that has been uncovered by the loss of peri-implant bone [13,14,15]. Following the machining, the surface of the implant is treated with chemical reagents to eliminate possible traces of bacteria or microorganisms [15,16,17,18].

One of the most commonly used treatments for these complications is the use of hydrogen peroxide (H_2_O_2_), using a concentration ranging between 3% and 5%. The bactericidal mechanism by which bacteria proliferation is altered consists of releasing bactericidal oxygen ions, without altering the metallurgical properties of titanium or the soft tissues [19,20,21,22]. However, the main drawback of this product is the application form, as well as its dosage, as the compound is liquid and is difficult to apply on the desired area.

Another antiseptic compound that has interesting properties that are applicable to peri-implant therapy is hypochlorous acid (HClO). This acid is a potent antimicrobial agent with oxidative effect, widely used in clinical medicine for infection control and wound repair. In vivo, it is synthesized by the immune system (neutrophils and macrophages) during antigen phagocytosis [23,24,25]. Several studies have shown that this compound has several properties, such as disinfectant, anti-inflammatory and regenerative properties [26,27].

Figure 1 shows a clinical case of the implantoplasty procedure and the application of these chemical reagents in peri-implant decontamination therapy. H_2_O_2_ was selected based on its higher oxygen concentration and for being able to decompose faster than carbamide peroxide, due to the high reactivity of its molecules that allows it to dissociate and release a series of free radicals with different electronic charges: hydrogen cations, hydroxyls, perhydroxyls, oxygen and water [28]. In this way, the product is able to exert its decontaminating antiseptic action on the treated area almost immediately [29]. Nevertheless, it is important to highlight that moderate concentrations of hydrogen peroxide can be used due to its strong oxidizing capacity and its ability to necrotize soft tissues [29,30,31,32].

The biosafety of this treatment is enhanced by the body’s own enzymatic capacity to neutralize possible residual amounts of H_2_O_2_, that is based on the existence of a catalase in our organism capable of degrading the product [29].

HClO is the active component present in sodium hypochlorite (bleach) that is applied to the area to be treated for short periods of time (2 min). It does not present the undesired side effects of dental staining attributed to other effective antiseptics, such as chlorhexidine [30,31,32,33,34,35,36]. The fact that HClO is also an endogenous substance synthesized by the immune system during phagocytosis [26] seems to play logically in favor of the biosafety of the compound [27].

In a recent work, Alovisi et al. [36] not only studied which treatment was the best disinfection system, but also whether the titanium surface could have good properties for cellular activity after the studied treatments. For this purpose, the study presented several disinfection methods in vitro on biofilm-coated machined and acid etched commercial pure grade 4 titanium, where the samples were infected with a vial of polymicrobial biofilm to simulate peri-implantitis. The treatments were carried out through immersion in glycine for 1 min; a local delivered triple paste antibiotic composed of a gel mixture with ciprofloxacin, metronidazole and clarithromycin for 1 h; and a combination of both. The combination of the two treatments showed the most relevant bactericidal effects capable to achieve a level of biocompatibility sufficient to allow cell growth.

In the current work, the effect of a disinfection treatment on the surface of a titanium dental implant after being subjected to machining to remove biofilm is determined. The influence on the titanium topography and the cytotoxicity and microbiological behavior when the treatment was performed with HClO or with HClO and H_2_O_2_ is determined. These treatments are usual after implantoplasty, since the bactericidal capacity of this solution is very high. However, the subsequent behavior of the treated surface, the changes in roughness and the behavior in the face of human cell cytocompatibility and its microbiological behavior have not been studied. We consider that it is important to understand the disinfectant effect produced at the moment of the application of chemical agents, as well as how it affects the surface for re-osseointegration and possible future bacterial colonization.

The first study hypothesis of the present in vitro study was to determine the effect of the disinfection treatments (HClO and HClO + H_2_O_2_) on the roughness of the titanium. The second study hypothesis was to study the changes in the cytotoxicity with osteoblastic cells and bacterial culture.

The null hypothesis was that the disinfection of titanium (HClO and HClO + H_2_O_2_) does not affect the surface of the dental implant and their future biological and microbiological behaviors. 

## 2. Materials and Methods

### 2.1. Materials

Seventy-five cylindrical specimens manufactured from commercially pure titanium grade 3 (Klockner Dental Implants, Escaldes Engordany, Andorra) with dimensions: 5 mm diameter and 2 mm width were cut, and three different surface conditions were studied after the applied the implantoplasty. This grade of titanium is very common in dental implants. 

Implantoplasty of the discs was carried out by the same investigator (EPG) (Figure 1). Using a GENTLEsilence LUX 8000B turbine (KaVo Dental GmbH, Biberach an der Riß, Germany) under constant irrigation, the surface was sequentially modified with a fine-grained tungsten carbide bur (reference H37931. 018 followed H37UF and H37931023), (Brasseler, KOMET; GmbH & Co., KG, Lemgo, Germany), a coarse-grained diamond polisher (Rug-byno. 9608.314.030 KOMET; GmbH & Co., KG, Lemgo, Germany) and a fine-grained silicon carbide polisher Arkansas stone and finishing amalgam (order no. 9618.314.553 KOMET; GmbH & Co., KG, Lemgo, Germany), as shown in Figure 1b. For each drill, a chemical treatment was applied (HClO, HClO + H_2_O_2_), for 5 s or 1 min (Figure 1c).

Figure 2 shows the chemical treatment after implantoplasty. Figure 2a shows the treatment with hydrogen peroxide gel, in which a bubble is produced by the reaction of hydrogen peroxide decomposition in oxygen and water. Figure 2b shows the hypochlorous acid treatment and Figure 2c shows the result of the implantoplasty with the chemical disinfection treatment 3 weeks later.

A general flowchart of the in vitro study is shown in Figure 3. These studies replicate the in vivo procedures explained above, in order to study the cytotoxicity and the microbiological behavior. Briefly, titanium discs were mechanically modified through implantoplasty. Then, the discs were sterilized using autoclave at 121 °C for 30 min. Finally, the discs were treated with HClO or HClO plus H_2_O_2,_ and the cellular and bacterial behaviors were analyzed.

### 2.2. Characterization of the Surfaces

Quantitative surface roughness was measured using a white light interferometer microscope (Wyko NT1100, Veeco Instruments Inc., Plainview, NY, USA) and proprietary software (Vison32, Veeco Instruments Inc., USA). The measurem ents were carried out in ten samples to characterize the average roughness (Ra), that represents the mean height of the peaks indicated by the arithmetic average of the absolute values of all points of the profile, and the real surface area (Ar) was larger than the nominal area (70.7 mm^2^) due to the surface roughness.

### 2.3. Cytotoxicity Assay

The cytotoxicity test consists of evaluating the percentage of cell survival of a known cell line when exposed to a medium that has been in contact with a given material. In this case, an indirect contact cytotoxicity test was performed according to the guidelines specified in the UNE EN ISO 10993-5 “Biological evaluation of medical devices”, part 5 entitled “In vitro cytotoxicity tests” [37]. To quantify cytotoxicity, the cell survival index was calculated, which indicates cytotoxicity if it is less than 70%.

The cytotoxicity of two treatments used in dentistry was evaluated: HClO and H_2_O_2_ + HClO; a phosphate buffered solution (PBS) was used as a control solution. Cytotoxicity was assessed by direct exposure determination.

All conditions studied in the cytotoxicity assay were performed in triplicate (n = 3), the samples studied are as follows:-Study samples: SAOS-2 cells cultured on discs with different treatments explained above at two time periods (5 s and 1 min);-Control treatment: SAOS-2 cells cultured on untreated discs (washed with PBS to simulate sample conditions);-Negative control: SAOS-2 cells seeded directly on the plate.

Sample handling was performed aseptically throughout the assay.

In accordance with the standard, SAOS-2 cells (ATCC, HTB-85) were used. Cells were stored using dimethyl sulfoxide (DMSO) as cryo-preservative at −180 °C (liquid nitrogen) and assayed bimonthly to verify the absence of mycoplasma contamination.

Cells were cultured in a humidity-controlled incubator at a temperature of 37 °C and with a 5% CO_2_ supply. As recommended by the manufacturer, they were cultured in McCoy’s 5A medium (Fisher Scientific, Waltham, MA, USA) supplemented with 10% fetal bovine serum (FBS, Sigma, Barcelona, Spain), 1% sodium pyruvate (Sigma-Aldrich, Barcelona, Spain) and 1% penicillin-streptomycin (Fisher Scientific). The medium was stored at 4 °C and the supplements at −20 °C.

The direct exposure test was performed according to section 8.3 of ISO 10993-5. For this, the discs were incubated with two treatments (HClO and H_2_O_2_ + HClO), using PBS as the control treatment, for 5 s and for 1 min. The treated discs were then transferred to 48-well plates and washed with culture medium. Cells were then seeded onto the discs at a density of 5 × 104 cells/disc. Cells seeded in the PBS-treated well plate were used as a treatment control, and cells seeded directly onto the plate were used as a negative control.

Cells were observed by light microscopy to verify the correct morphology before seeding on the treated discs. Once the assay was completed, cell viability was assessed using the resazurin sodium salt protocol (Sigma) following the manufacturer’s recommendations. This assay uses a blue product (resazurin) that is reduced by the cells and forms a product of different coloration (resorufin), that can be analyzed by absorbance. For this purpose, a dilution with complete resazurin medium (initial concentration 5 mg/mL) was prepared until a final concentration of 10 µg/mL was obtained, and the cells were cultured with the resazurin medium for 3 h in the incubator at 37 °C and 5% CO_2_. Subsequently, absorbance was measured at 570 nm using a plate reader. This procedure was performed 24 and 72 h after culture on the discs.

### 2.4. Microbiological Response Assay

The bacterial bioactivity assay (BacTiter-Glo™ Microbial Cell Viability Assay, Madison, WI, USA) was performed using the bacterial strains *S. gordonii* and *S. oralis*. The strains were chosen due to their frequent presence in the oral cavity, as they play an important role in the production of dental biofilm and alkalinization of the mouth. The strains were obtained from the Spanish Type Culture Collection (CECT, University of Valencia, Spain). Three replicates (n = 3) per treatment were used for the assay.

Upon autoclaving the samples at 121 °C for 30 min, the treatments were applied. The two treatments to be studied (HClO and H_2_O_2_ + HClO) were performed and PBS was used as a control. The incubation time of the treatment was analyzed by incubating the discs in the treatment for 5 s and also for 1 min. Then, each sample was washed three times with 400 µL of BHI (brain heart infusion broth) bacterial medium and finally seeded with 350 µL of bacterial solution and incubated for 24 h at 37 °C.

Following 24 h of incubation, the BacTiter-Glo™ microbial cell viability assay was performed based on the manufacturer’s instructions. For this, the discs were transferred to a 48-well plate and the discs were washed with 150 µL of phosphate buffered solution (PBS). Then, 150 µL of reagent (ATP reagent), previously diluted to 50%, was added. It was kept in the incubator under agitation for 5 min. Following the incubation time, 95 μL from each well were transferred into an opaque 96-well plate and finally luminescence was measured in an Infinite^®^ 200 PRO multimode absorbance microplate reader (TECAN).

### 2.5. Statistical Analysis

Statistically significant differences among the test groups for histometry were assessed using statistical software (MinitabTM 13.1, Minitab Inc., San Jose, CA, USA). ANOVA tables with a multiple comparison Fisher test were calculated. The level of significance was established at *p*-value < 0.01.

## 3. Results

The roughness measurements are shown in Table 1. Both peak-to-peak distances (Ra) and peak-to-valley heights (Rz) increase with the chemical treatments. Ra and Rz reveal that the roughness increase is even more evident when both treatments are combined.

Figure 4 shows the surfaces observed by SEM of the control titanium, titanium with HClO treatment and titanium treated with HClO and H_2_O_2_. As observed in the images, the effect of hydrogen peroxide is the one that contributes to the roughness of the titanium surface in a more important way.

### 3.1. Cytotoxicity Assay

The cytotoxicity assay was performed at 24 and 72 h to evaluate cell survival. For this purpose, SAOS-2 osteoblastic cells were cultured in the different treated discs.

Following the regulations, a cell survival rate lower than 70% was considered as cytotoxic. To ensure a correct study, the cells were adhered to the tissue culture polystyrene and presented the expected morphology before incubation on the discs.

As can be seen in Figure 5, cells cultured in the experimental discs present similar rates of survival at 24 h, having similar values to the control (PBS) both at 5 s and 1 min of incubation (Figure 5A). Moreover, Figure 5B shows similar results for the HClO treatment and the control at 72 h, whereas the combinatorial treatment is cytotoxic since cell viability decreases to 24.68% (at 5 s) and 23.57% (at 1 min). As is well known, the decomposition of hydrogen peroxide causes water and oxygen, the latter causing the oxidation of bacteria, but also of cells and proteins in the physiological environment.

### 3.2. Bacterial Viability Analysis

Bacterial viability analysis was performed to determine the bactericidal capacity of the treatments. For this purpose, two common oral strains were used, Streptococcus gordonii and Streptococcus oralis, to analyze its ATP activity. As can be seen in Figure 6, both strains follow the same trend since the treatment with HClO alone seems to maintain or decrease bacterial viability while treatment combining H_2_O_2_ and HClO increases the percentage of viability. This change is significant in the case of *S. gordonii*, where the values increase to 1599% (at 5 s of incubation) and 655% (at 1 min of incubation).

## 4. Discussion

Implantoplasty treatments cause an increase in residual stresses on the titanium surface due to the machining processes aimed at removing biofilm. The works of Lozano et al. and Toledano et al. [11,12,14] determined how these deformations caused changes in the surface behavior, such as an increase in the corrosion rate, an increase in the release of metal ions to the medium as well as an increase in the hardness values. In other words, implantoplasty generates a less passive surface than that of titanium that has not undergone implantoplasty. This fact triggers the appearance of a more vulnerable titanium surface, that can be attacked with hypochlorous acids, and especially the combination of this acids with H_2_O_2_ generating an increase in the roughness of the surface of the dental implant.

It is well demonstrated that this chemical treatment on the surface of the dental implant and surrounding tissues causes the death of bacteria and microorganisms present due to the high concentration of oxygen that causes the oxidation of bacteria. However, the slight increase in roughness facilitates new bacterial adhesion and bacterial plaque formation. Gil et al. studied the influence of roughness on bacterial adhesion on titanium and the values obtained by acid treatment with HClO and H_2_O_2_ caused an optimal roughness for new bacteria to adhere and proliferate [38,39]. The effects of the oxygen concentrations disappear after several hours of the treatment and the bactericidal effect on the titanium surface subjected to implantoplasty, as well as on the surrounding tissues, is inhibited. Caution should be exercised on the concentration of H_2_O_2_ as it can cause tissue necrosis. For this reason, the use of 6% H_2_O_2_ is recommended.

Hydrogen peroxide is responsible for the bactericidal effects observed in biological systems, such as the growth inhibition of one bacterial species by another and the killing of invading microorganisms by activated phagocytic cells. Hydrogen peroxide reacts very fast. The disinfection mechanism of hydrogen peroxide is based on the release of free oxygen radicals: H_2_O_2_ → H_2_O + O_2_.

Bacteria are decomposed by free oxygen radicals, and only water remains. Free radicals have both oxidizing and disinfecting abilities. The oxidation of the various molecules that constitute microorganisms will lead to significant disruptions in structure/function and the loss of viability or infectivity. Despite this generalization, liquid preparations, formulations and the gas form of hydrogen peroxide can show remarkable differences in their antimicrobial effects, such as their attack on proteins, nucleic acids and lipids [40].

However, the roughness obtained in these treatments is small for good adhesion and proliferation of osteoblastic cells. This effect of optimal topography for cellular activity is well explained by the research of Lausma and Kasemo, where their studies revealed that Ra values below 1 micrometer were not suitable for osseointegration [41,42].

It has been proven that when implantoplasty treatments are performed, local inflammation is provoked as a response of the immune system. This causes a decrease in oxygen concentration that kills aerobic bacteria. This decrease in oxygen causes a reduction of the titanium oxide layer to pure titanium. This protective mechanism means that when the inflammation is reduced the oxygen ingress causes an oxidation of the titanium, producing oxides with different stoichiometries that are not compatible [43]. Different antibiotic systems have been tested for the fight against biofilm, but have not been very successful due to their short-lived action, as well as the fact that bacteria are protected from these new drugs [44,45,46,47,48,49,50,51,52,53,54,55,56,57,58].

These results make very important the search for a chemical solution that allows the titanium to be re-passivated once the implant has been performed in order to avoid cytotoxicity, and to achieve a surface state of the implant that allows re-osseointegration while inhibiting bacterial colonization. Implantoplasty is a solution to avoid the presence of biofilm, but causes many negative effects, such as: release of particles of different sizes into the environment, titanium surfaces with high residual stress that favor titanium degradation: corrosion, release of ions, as well as a loss of mechanical properties. Biomaterials researchers are called to create a dental implant with a bactericidal or at least bacteriostatic capacity that will allow the good long-term behavior of dental implants and reduce peri-implantitis.

Implantoplasty is a technique widely used all over the world, and the disinfectant treatments used in the mouth must be compatible with human tissues in order not to necrotize them. The most common treatments are those we have studied in this contribution with hypochlorous acid and hydrogen peroxide, due to their disinfectant capacity thanks to their oxidative power [59,60]. Likewise, other agents, such as calcium hydroxide with a very basic pH have also been studied, as well as ozone treatments. These latter treatments are effective in bacterial elimination, although they are very aggressive to the tissues and can cause the death of the soft tissues [59,60]. For this reason, treatment with hypochlorous acid and hydrogen peroxide is increasingly used in dental clinics.

The problem that arises with these treatments is that although they instantly have a very effective bactericidal capacity, they stop having it after a short time, approximately 2–3 days. Sometimes, the clinician offers antibiotics for a week to extend the bactericidal effect. However, the effect of these chemical agents causes surface defects in titanium and local concentrations of oxidizing components that inhibit the migration of osteoblast progenitor proteins, as we have seen in the cytotoxicity results of SaOS-2 cells in this study. Furthermore, roughness favors the colonization of bacteria, as we have seen in this study, forming colonies and biofilms more easily, and these are protected by the rough topography of the surface [61,62]. It is therefore important that disinfection protocols are effective at the time, avoiding any negative effect in the longer term stability of the implant, especially at the biological level. Advances must be made to establish a protocol in which disinfection occurs and generates on the surface a re-passivation of the titanium. Of increased significance would be the development of a surface that presents a certain bioactivity for the formation of bone tissue in the body of the implant and a soft tissue in the neck of the implant for the creation of a biological seal that prevents the filtration of bacteria.

From the results in the present work, we have been able to demonstrate that HClO and H_2_O_2_ treatment affects the roughness of the titanium discs subjected to implantoplasty. The roughness favors bacteria proliferation in the strains and the effect of the H_2_O_2_ increase in the cytotoxicity with osteoblastic cells. The null hypothesis is therefore rejected.

## 5. Conclusions

The treatment after implantoplasty by H_2_O_2_ and HClO causes an increase in the roughness of the titanium disc with implantoplasty, showing statistically significant differences. Cell survival at 24 h in the treatments is similar to that of the control, while at 72 h of culture, the treatment combining H_2_O_2_ and HClO is significantly cytotoxic. Bacterial viability follows the same trend in the two strains (*Streptococcus gordonii* and *Streptococcus oralis*), the increase in viability is significantly much higher in the first strain, in the case of the treatment combining HClO and H_2_O_2_.

## Figures and Tables

**Figure 1 materials-16-02953-f001:**
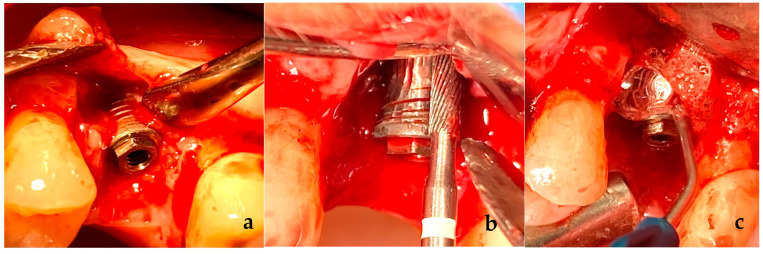
(**a**). Intraoral view of the bone defect after the surgical access. (**b**). Implantoplasty. Mechanical decontamination with diamond burs and irrigation with HClO. (**c**). Application of 6% H_2_O_2_, leaving it to act for 1 min, and then washing with HClO as an additional decontaminating agent.

**Figure 2 materials-16-02953-f002:**
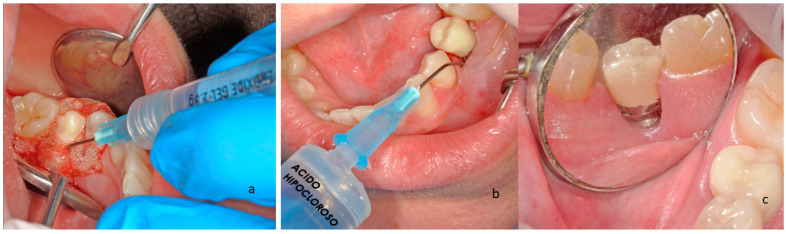
(**a**). Hydrogen peroxide treatment on the titanium surface after the mechanical modification. (**b**). Hypochlorous acid treatment. (**c**). The implant and the soft tissues after three weeks.

**Figure 3 materials-16-02953-f003:**
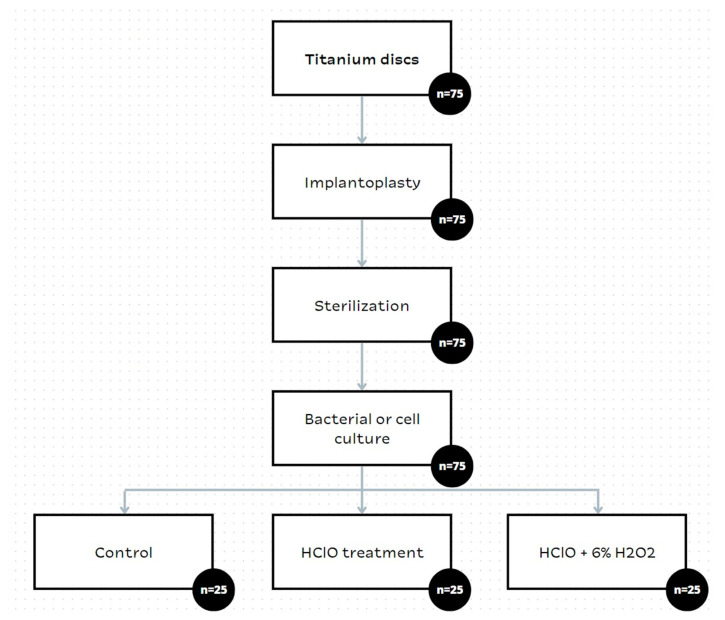
Scheme of the research of this study. The number represents to the total number of studied discs.

**Figure 4 materials-16-02953-f004:**
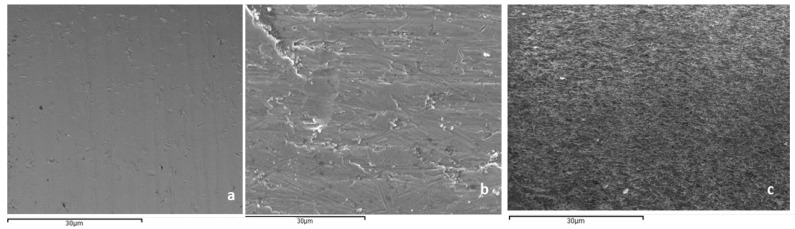
Surfaces with different chemical treatments. (**a**). Control. (**b**). Treated with hypochlorous acid. (**c**). Treated with hypochlorous acid and hydrogen peroxide.

**Figure 5 materials-16-02953-f005:**
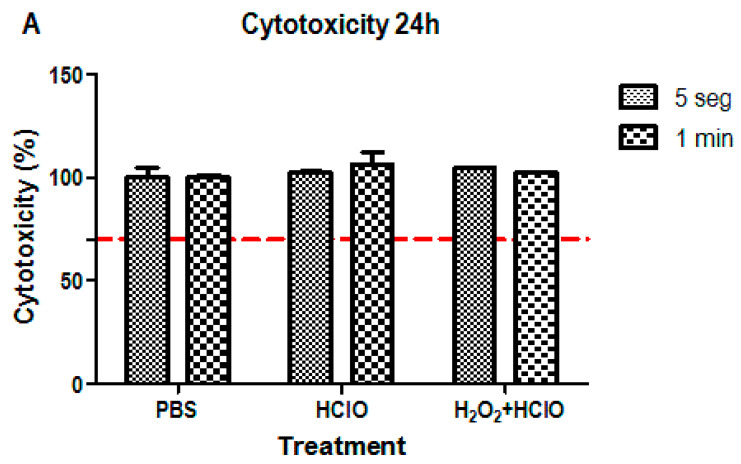
Cytotoxicity levels of SAOS-2 cells. Cultures performed at 24 h (**A**) and 72 h (**B**) incubating the discs in the treatment solution for 5 s and for 1 min. The treatments used were hypochlorous acid (HClO) and hydrogen peroxide (H_2_O_2_) + HClO, using a PBS treatment as control. * *p* < 0.05. The red line indicates the minimum level for cytocompatibility.

**Figure 6 materials-16-02953-f006:**
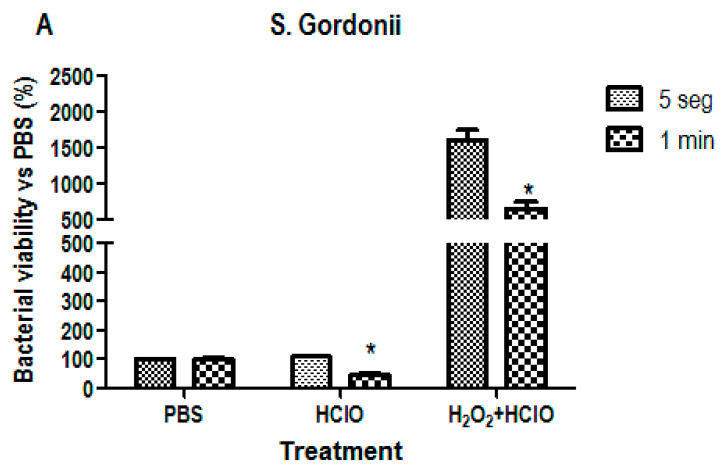
Bacterial viability analysis. Cultures performed on Streptococcus gordonii (**A**) and Streptococcus oralis (**B**) by incubating the discs in the treatment solution for 5 s and for 1 min. The treatments used were hypochlorous acid (HClO) and hydrogen peroxide (H_2_O_2_) + HClO, using a PBS treatment as control. * *p* < 0.05.

**Table 1 materials-16-02953-t001:** Roughness for each dental implant studied. * Means differences statistically significant with *p* < 0.01.

Implant	R_a_ (μm)	R_z_ (μm)
Titanium-control	0.33 ± 0.12	3.10 ± 0.69
Titanium + HClO	0.41 ± 0.11	3.20 ± 0.34
Titanium + HClO + H_2_O_2_	0.68 ± 0.23 *	3.92 ± 0.36 *

## Data Availability

The authors can provide details of the research requesting by letter and commenting on their needs.

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
