# Peer review of "Effects of Hypochlorous Acid and Hydrogen Peroxide Treatment on Bacterial Disinfection Treatments in Implantoplasty Procedures"

_materials, 2023, doi:10.3390/ma16082953_

Round 1

Reviewer 1 Report

Introduction: 

- Line 64: ''We illustrate in Figure 1, by means of a clinical case the implantoplasty procedure...'' Avoid the use of ''we''. Reformulate in ''Figure 1 shows a clinical case....''

Carefully check all the text (example line 105 ''In this research we intend...'') and correct the issue throughout the text.

- The state of the art on the bacterial disinfection to eliminate the biofilm is currently missing. Add some sentences described the state of the art on the topic. It will help to both allow the reader to deepening the knowledges on the topic and understand the rational of the study.

For this propose, discuss and cite the following recently published article on the topic.

 Alovisi, M.; Carossa, M.; Mandras, N.; Roana, J.; Costalonga, M.; Cavallo, L.; Pira, E.; Putzu, M.G.; Bosio, D.; Roato, I.; Mussano, F.; Scotti, N. Disinfection and Biocompatibility of Titanium Surfaces Treated with Glycine Powder Airflow and Triple Antibiotic Mixture: An In Vitro Study. Materials 202215, 4850. https://doi.org/10.3390/ma15144850

- Add the study hypothesis after the aim of the study

Materials and methods 

line 105: ''...and three different surface conditions were studied after the applied the implantoplasty.'' Remove ''the''

- I had some confusions about the procedures.  in paragraph ''2.1 Materials'' you described that ''25 discs treated with HClO '' and 25 discks were treated with HClO + H2O2. In the same paragraph you described that the disks were then autoclaved. Later, it appears that, after the autoclaved, you inserted the bacterial, cells, etc. but it doesn't make sense to apply this methods after the disinfection protocols. I do think that it is just a mistake of presenting the procedures, therefore please reformulate the paragraph to make the procedure and the timing of each process more clear.

- Following the previous point, please add a flow chart of the study at the beginning of the M&M section.

Discussion:

- Discuss if the study hypothesis were accepted or rejected accordingly to the result of the study

Author Response

REVIEWER 1

Dear Reviewer,

Thanks for taking the time to review our manuscript and suggest to us to improve our work by providing a lot more detail. We have done so, and we are now submitting a manuscript that not only addresses the points you specifically raised but also many others that we have considered in order to deliver what we think is a much improved version of our work. This version includes more paragraphs, new figures, English grammar revisions in all main sections, new references. Thanks a lot and happy new year. We are looking forward to your comments.

Sincerely,

Francisco-Javier Gil Mur

Introduction:

- Line 64: ''We illustrate in Figure 1, by means of a clinical case the implantoplasty procedure...'' Avoid the use of ''we''. Reformulate in ''Figure 1 shows a clinical case....''

DONE

Carefully check all the text (example line 105 ''In this research we intend...'') and correct the issue throughout the text.

REVISED

- The state of the art on the bacterial disinfection to eliminate the biofilm is currently missing. Add some sentences described the state of the art on the topic. It will help to both allow the reader to deepening the knowledges on the topic and understand the rational of the study.

For this propose, discuss and cite the following recently published article on the topic.

 Alovisi, M.; Carossa, M.; Mandras, N.; Roana, J.; Costalonga, M.; Cavallo, L.; Pira, E.; Putzu, M.G.; Bosio, D.; Roato, I.; Mussano, F.; Scotti, N. Disinfection and Biocompatibility of Titanium Surfaces Treated with Glycine Powder Airflow and Triple Antibiotic Mixture: An In Vitro Study. Materials 2022, 15, 4850. https://doi.org/10.3390/ma15144850.

A new paragraphs and new reference have been introduced according to the reviewer. Thank you for your help.

- Add the study hypothesis after the aim of the study

The authors have introduced two hypothesis and one null-hypothesis in the last paragraph in the introduction.

Materials and methods

line 105: ''...and three different surface conditions were studied after the applied the implantoplasty.'' Remove ''the''

Done

- I had some confusions about the procedures.  in paragraph ''2.1 Materials'' you described that ''25 discs treated with HClO '' and 25 discks were treated with HClO + H2O2. In the same paragraph you described that the disks were then autoclaved. Later, it appears that, after the autoclaved, you inserted the bacterial, cells, etc. but it doesn't make sense to apply this methods after the disinfection protocols. I do think that it is just a mistake of presenting the procedures, therefore please reformulate the paragraph to make the procedure and the timing of each process more clear.

This aspect has been clarified.

- Following the previous point, please add a flow chart of the study at the beginning of the M&M section.

Discussion:

- Discuss if the study hypothesis were accepted or rejected accordingly to the result of the study

Done according to the comment of the reviewer

Reviewer 2 Report

This study is interesting and useful for other researchers, and can be considered for publication in Materials after a minor revision according to the following comments:

1- The novelty of the work should be highlighted.

2- Results and discussion are good, but what does this research add to the subject area compared to other published cases?

3- The benefits of using H2O2 in this study should be mentioned.

4- Conclusions are consistent with the evidence, and the references are appropriate.

5- A few typos should be corrected. For example: line 186: "CO2" should be "CO2".6.  The references are appropriate.

Author Response

REVIEWER 2

Dear Reviewer,

Thanks for taking the time to review our manuscript and suggest to us to improve our work by providing a lot more detail. We have done so, and we are now submitting a manuscript that not only addresses the points you specifically raised but also many others that we have considered in order to deliver what we think is a much improved version of our work. This version includes more paragraphs, new figures, English grammar revisions in all main sections, new references. Thanks a lot and happy new year. We are looking forward to your comments.

Sincerely,

Francisco-Javier Gil Mur

This study is interesting and useful for other researchers, and can be considered for publication in Materials after a minor revision according to the following comments:

1- The novelty of the work should be highlighted.

The novelty of this research has been added in the introduction according to the reviewer.

2- Results and discussion are good, but what does this research add to the subject area compared to other published cases?

In the discussion has been introduced aspects of different methods and their problems. The authors have introduced new paragraphs and references.

3- The benefits of using H2O2 in this study should be mentioned.

New paragraphs and references about the benefits of using H2O2 have been introduced in the text according to the reviewer

4- Conclusions are consistent with the evidence, and the references are appropriate.

Thank you

5- A few typos should be corrected. For example: line 186: "CO2" should be "CO2".6.  The references are appropriate.

Done.

Reviewer 3 Report

Dear authors,

The manuscript titled: „ Effects of hypochlorous acid and hydrogen peroxide treatment  in bacterial disinfection treatments in implantoplasty processes“ is interesting manuscript with main aim to evaluate one of the  chemical treatments based on hypochlorous acid (HClO) and hydrogen peroxide (H2O2) in combination with implantoplasty.

However I have many questions/suggestions to clarify. Please look at my notes.

There are some grammar mistakes in manuscript so please check once more the whole text.

In abstract You should emphasize that this is in vitro study because this is a little bit confusing.

The Introduction section well sets up the main topic and introduce the development of the manuscript. However, the null hypothesis is not expressed. The null hypothesis statement must precisely identify the variables assessed through statistical analysis. Please add its statement in the last part of the Introduction section.

Why did You use grade 3 titanium? Why did You include clinical picture (Fig. 1) in this manuscript? There is a obvious lack of figures in which You should show whole procedure step by step so readers can follow study.

S. Gordonii and S. Oralis are not the best choice for imitating periimplantitis, why did You use these bacteria?

What type of statistical analysis was made? There is lack of this informations in manuscript.

Presentation of results is very poor, with lack of many informations.

Discussion is too short with general scientific facts and referring on references that are out of date.

Manuscript has serious flaws regarding methodology, statistical analysis, presentation of results and in the end discussion part.

Author Response

REVIEWER 3

Dear Reviewer,

Thanks for taking the time to review our manuscript and suggest to us to improve our work by providing a lot more detail. We have done so, and we are now submitting a manuscript that not only addresses the points you specifically raised but also many others that we have considered in order to deliver what we think is a much improved version of our work. This version includes more paragraphs, new figures, English grammar revisions in all main sections, new references. Thanks a lot and happy new year. We are looking forward to your comments.

Sincerely,

Francisco-Javier Gil Mur

Dear authors,

The manuscript titled: „ Effects of hypochlorous acid and hydrogen peroxide treatment  in bacterial disinfection treatments in implantoplasty processes“ is interesting manuscript with main aim to evaluate one of the  chemical treatments based on hypochlorous acid (HClO) and hydrogen peroxide (H2O2) in combination with implantoplasty.

However I have many questions/suggestions to clarify. Please look at my notes.

There are some grammar mistakes in manuscript so please check once more the whole text.

The text has been revised by native English

In abstract You should emphasize that this is in vitro study because this is a little bit confusing.

The abstract has been clarified.

The Introduction section well sets up the main topic and introduce the development of the manuscript. However, the null hypothesis is not expressed. The null hypothesis statement must precisely identify the variables assessed through statistical analysis. Please add its statement in the last part of the Introduction section.

The authors have introduced two hypothesis and one null-hypothesis in the last paragraph in the introduction.

Why did You use grade 3 titanium? Why did You include clinical picture (Fig. 1) in this manuscript? There is a obvious lack of figures in which You should show whole procedure step by step so readers can follow study.

Grade 3 of commercially pure titanium has been chosen as it is very common in the manufacture of dental implants. This comment has been introduced in the text.

Also, the authors have added a flow chart of the methodology to help readers in accordance with the reviewer's commentary.

  1. Gordoniiand S. Oralisare not the best choice for imitating periimplantitis, why did You use these bacteria?

The reason is that in a previous article already published the reviewers advised us to use Gram+ and Gram- bacteria for future studies and that is why we changed Lactobacillus Salivarius to Gordonii. The microbiologists told us that it was suitable for the study. Although, as the reviewer says, there may be other bacteria that play an important role, there are so many species found in the biofilm that we chose this one on the advice of the microbiologists. In any case, it seems that roughness favors bacterial growth in general.

What type of statistical analysis was made? There is lack of this informations in manuscript.

The statistical analysis has been introduced in Materials and Methods according to the reviewer

Presentation of results is very poor, with lack of many informations.

The results have been improved and statistical analysis

Discussion is too short with general scientific facts and referring on references that are out of date.

The discussion has been improved, new paragraphs, references and comparison with other authors.

Manuscript has serious flaws regarding methodology, statistical analysis, presentation of results and in the end discussion part.

The authors have improved all the aspects of the contribution

Reviewer 4 Report

- Line 16: Please replace the term "mechanization" with the correct term. 
- Please use the definition for "peri-implantitis" according to the 2017 AAP Workshop; "Peri-implant diseases and conditions: Consensus report of workgroup 4 of the 2017 World Workshop on the Classification of Periodontal and Peri-Implant Diseases and Conditions".

- Line 47: Please correct the term "machining of the implant surface" with the correct one; "mechanically modifying the implant surface topography" 

- In figure 1b, we can see the incorrect placement of an external hex implant where clearly the machined surface was placed in a subcrestal way. The crestal bone, in such cases, will physiologically remodel where the rough surface of the implant exists, which it might be misinterpret as "peri-implantitis". Figures 2c and 2d show bone loss more than the expected remodelling of the crestal bone around the rough surface of the implant, suggesting "peri-implant disease". In order to confirm the amount of bone loss, a peri-apical xray or a vertical bitewing should be used from the day the final crown was placed; that should be the baseline. The presence or not of the palatal plate should be evaluated in order to proceed with the correct treatment plan, which it might combine implantoplasty along with guided bone regeneration, depending of the presence (or not) of the palatal/lingual plate, the interproximal bone level and the shape of the defect. Unfortunately, there is no black or white when the correct treatment of peri-implantitis is discussed and decided. 

- Figure 1a looks like a different case than the one shown in figures 1b-1d. It would be good to show the same case in your figures so the readers won't be confused. 

- Please rephrase the purpose of your study in lines 105-109.

- Was any power analysis performed in order to have 25 samples per group?  

- What is the purpose to increase the roughness on a surface where was treated before with implantoplasty? In addition, what is the purpose to increase the survival of bacterial pathogens on the same surface? 

- In order to make more sense, your control group should have included specimens where you have performed implantoplasty, prophy jet with glycine powder and saline rinse, or at least implantoplasty with saline rinse. 

- The overall purpose of the study is a proof of concept that the treatments used on your study groups don't work and should not be performed by clinicians to treat peri-implantitis? 

- The message that you are trying to pass from the results of you study is not clear and the readers might be confused. 

Author Response

REVIEWER 4

Dear Reviewer,

Thanks for taking the time to review our manuscript and suggest to us to improve our work by providing a lot more detail. We have done so, and we are now submitting a manuscript that not only addresses the points you specifically raised but also many others that we have considered in order to deliver what we think is a much improved version of our work. This version includes more paragraphs, new figures, English grammar revisions in all main sections, new references. Thanks a lot and happy new year. We are looking forward to your comments.

Sincerely,

Francisco-Javier Gil Mur

Line 16: Please replace the term "mechanization" with the correct term.

Done

- Please use the definition for "peri-implantitis" according to the 2017 AAP Workshop; "Peri-implant diseases and conditions: Consensus report of workgroup 4 of the 2017 World Workshop on the Classification of Periodontal and Peri-Implant Diseases and Conditions".

Done

- Line 47: Please correct the term "machining of the implant surface" with the correct one; "mechanically modifying the implant surface topography"

Done

- In figure 1b, we can see the incorrect placement of an external hex implant where clearly the machined surface was placed in a subcrestal way. The crestal bone, in such cases, will physiologically remodel where the rough surface of the implant exists, which it might be misinterpret as "peri-implantitis". Figures 2c and 2d show bone loss more than the expected remodelling of the crestal bone around the rough surface of the implant, suggesting "peri-implant disease". In order to confirm the amount of bone loss, a peri-apical xray or a vertical bitewing should be used from the day the final crown was placed; that should be the baseline. The presence or not of the palatal plate should be evaluated in order to proceed with the correct treatment plan, which it might combine implantoplasty along with guided bone regeneration, depending of the presence (or not) of the palatal/lingual plate, the interproximal bone level and the shape of the defect. Unfortunately, there is no black or white when the correct treatment of peri-implantitis is discussed and decided.

I am very grateful to the reviewer for his comments on the clinical aspects of the images. The authors only wanted to illustrate to the non-clinical readers of Materials magazine, the mechanization processes of dental implants, so that they would have a clear idea of the mechanical processes and the problems that are generated in the treatment of implantoplasty. The truth is that we did not go into the details of dental implant placement, nor did we intend to delve into the more clinical aspects, but especially into the part of materials. We are very grateful to the reviewer for the explanations, and we have changed some of the images so as not to confuse the readers. I reiterate my thanks to his comment.

- Figure 1a looks like a different case than the one shown in figures 1b-1d. It would be good to show the same case in your figures so the readers won't be confused.

The reviewer is right. Thank you very much for your input as it might confuse readers. Figure 1a has been removed.

- Please rephrase the purpose of your study in lines 105-109.

Done

- Was any power analysis performed in order to have 25 samples per group?

If we carry out the study and the 25 samples are sufficient to be able to develop the study with statistical rigor.  

- What is the purpose to increase the roughness on a surface where was treated before with implantoplasty? In addition, what is the purpose to increase the survival of bacterial pathogens on the same surface?

No attempt has been made to increase roughness, but the chemical agents used for disinfection, especially oxygenated water, cause an increase in roughness. This topographical change is what generates the greatest bacterial adhesion in the cultures. The work aims to explain that the chemical disinfection process affects the topography, and this change has consequences on cytotoxicity and bacterial proliferation over time.

This fact has been clarified in the text.

- In order to make more sense, your control group should have included specimens where you have performed implantoplasty, prophy jet with glycine powder and saline rinse, or at least implantoplasty with saline rinse.

Yes, the control group has saline rinse after implantoplasty as indicated by the reviewer. It has been clarified in Materials and methods.

- The overall purpose of the study is a proof of concept that the treatments used on your study groups don't work and should not be performed by clinicians to treat peri-implantitis?

The goal is to show readers that disinfection treatments can remove bacteria and effectively clean the surface but can affect the topography of the titanium as well as the nature of the titanium surface which in the long term can again accelerate infection.

- The message that you are trying to pass from the results of you study is not clear and the readers might be confused.

The authors have added the hypotheses and null hypothesis, added a flowchart, improved the discussion of results and clarity in Materials and Methods. I believe that in this latest version the objective, originality and conclusions of the study are clearer as suggested by the reviewer. Thank you very much for your help.

Round 2

Reviewer 1 Report

Dear Authors,

thank you for addressing my points.

Author Response

Thank you for your help. 

Reviewer 3 Report

Dear Authors,

There are further grammar mistakes in manuscript so editing of English language is still required.

You added hypotheses but they are not in correlation with your work so I suggest to modify this.

You did not add any new figure in manuscript??? It is very important to show the whole procedure step by step, as it was suggested in previous review.

Author Response

REVIEWER 3

Thank you very much again for your important considerations. The authors have corrected all the suggestion according to the reviewer.

Dear Authors,

There are further grammar mistakes in manuscript so editing of English language is still required.

English has been revised

You added hypotheses but they are not in correlation with your work so I suggest to modify this.

The authors have modified the hypotheses

You did not add any new figure in manuscript??? It is very important to show the whole procedure step by step, as it was suggested in previous review.

The authors have introduced two new figures: disienfectation process and the surfaces observed by SEM for different treatments according to the comment of the reviewer. Thank you very much for this important suggestion.

Reviewer 4 Report

Extensive editing of English language and style are required. The manuscript has several mistakes. 

- Next to your study flowchart, please add some pictures from your actual in vitro study such as the disks, the decontamination process etc. It is essential for us and the reviewers to see how you actually did your study. 

- Please replace on the title of the paper the term "processes" to "procedures"

- Your study hypotheses don't correlate with what you actually did, please rephrase it so it shows exactly what you did. Please speak with a statistician to help you write the purpose of your study.  

- Line 16: please replace the term "machining" with the correct one "mechanical modification of the implant surface topography".

- Line 19: When you start a sentence, please don't use numbers (25) but words (twenty-five). Please correct similar mistakes on the manuscript. 

- Line 80; You repeat the same phrase twice, please correct it. 

- Line 80; "Scientific on  Consumer Products", please elaborate on that, does not make sense the way it is written.

- Line 84; "caustic power they present", please explain it properly, it does not make sense the way it is written. 

- Line 92; You are repeating the same phrase twice. Please correct it. 

- Line 103; please rephrase "were realized by glycine powder air-flow", it does not make sense the way it is written. 

- Line 157; it is a flowchart not a "scheme". Please rephrase. 

Author Response

REVIEWER 4

Thank you very much again for your important considerations. The authors have corrected all the suggestion according to the reviewer.

Extensive editing of English language and style are required. The manuscript has several mistakes. 

The authors have correct several mistakes and the style has been revised

- Next to your study flowchart, please add some pictures from your actual in vitro study such as the disks, the decontamination process etc. It is essential for us and the reviewers to see how you actually did your study. 

The authors have introduced two new figures: disienfectation process and the surfaces observed by SEM for different treatments according to the comment of the reviewer. Thank you very much for this important suggestion.

- Please replace on the title of the paper the term "processes" to "procedures"

Done

- Your study hypotheses don't correlate with what you actually did, please rephrase it so it shows exactly what you did. Please speak with a statistician to help you write the purpose of your study.  

- Line 16: please replace the term "machining" with the correct one "mechanical modification of the implant surface topography".

Done

- Line 19: When you start a sentence, please don't use numbers (25) but words (twenty-five). Please correct similar mistakes on the manuscript. 

Done

- Line 80; You repeat the same phrase twice, please correct it. 

The sentence has been re-written according to the reviewer-

- Line 80; "Scientific on  Consumer Products", please elaborate on that, does not make sense the way it is written.

The comment has been deleted.

- Line 84; "caustic power they present", please explain it properly, it does not make sense the way it is written. 

This expression has been changed and explained.

- Line 92; You are repeating the same phrase twice. Please correct it. 

Corrected

- Line 103; please rephrase "were realized by glycine powder air-flow", it does not make sense the way it is written. 

Done

- Line 157; it is a flowchart not a "scheme". Please rephrase. 

Done